# Conductometric Studies of Formation the Inclusion Complexes of Phenolic Acids with β-Cyclodextrin and 2-HP-β-Cyclodextrin in Aqueous Solutions

**DOI:** 10.3390/molecules28010292

**Published:** 2022-12-29

**Authors:** Zdzisław Kinart

**Affiliations:** Department of Physical Chemistry, Faculty of Chemistry, University of Lodz, Pomorska 163/165, 90-236 Lodz, Poland; zdzislaw.kinart@chemia.uni.lodz.pl

**Keywords:** electric conductivities, *β*-cyclodextrin, 2-HP-*β*-cyclodextrin, aqueous solutions of sodium salts of phenolic acids, complex constants, thermodynamic function

## Abstract

An attempt was made to evaluate the possibility of creating and assessing the stability of inclusion complexes of selected phenolic acids [*trans*-4-hydroxycinnamic acid (*trans*-p-coumaric acid), *trans*-3,4-dihydroxycinnamic acid (*trans*-caffeic acid), *trans*-4-hydroxy-3-methoxycinnamic acid, (*trans*-ferulic acid) and *trans*-3-phenylacrylic acid (*trans*-cinnamic acid)] with *β*-cyclodextrin and 2-HP-*β*-cyclodextrin in aqueous solutions in a wide temperature range 283.15 K–313.15 K. On the basis of the values of the limiting molar conductivity *(Λ_CDNaDod_*), calculated from the experimental data, the values of the formation constants and the thermodynamic functions of formation (standard enthalpy, entropy, and Gibs standard enthalpy) of the studied complexes were determined. It has been found that the stability of the studied complexes increases with lowering of the molar mass of cyclodextrin and lowering of the temperature.

## 1. Introduction

Cyclodextrins (CD), also called cycloamylases, cycloglucons, or Schardinger dextrins, are naturally occurring cyclic glucose oligosaccharides containing 6 (*α*-cyclodextrin), 7 (*β*-cyclodextrin) or 8 (*γ*-cyclodextrin) linked by an α-1.4 bond of sugar molecules in a ring [1]. Villiers obtained these compounds for the first time in 1891 by decomposing starch with Bacillus amylobacter bacteria. The structural structure of cyclodextrins was described by Schardinger in 1903. Currently, these compounds are obtained by hydrolysis of starch using the extracellular CD glucosyltransferase (CGT) enzyme produced by microorganisms, i.e., *Bacillus macerans*, *Bacillus circulans*, *Bacillus coagulans*, *Klebsiella pneumoniae* [1]. Cyclodextrins have the shape of toroidal rings whose internal diameter is equal to that of *α*-cyclodextrin = 4.5 Å, *β* = 7.0 Å, *γ* = 8.5 Å. Because of the different sizes of the cyclodextrin cavities, it is characterized by complex selectivity. Primary hydroxyl groups bonded to the carbon atom of glucose are located outside the ring, and the secondary hydroxyl groups linked to the glucose atoms C-2 and C-3 are located inside the torus. An important consequence of this structure is the fact that the inside of the ring is nonpolar (resulting from the presence of hydrophobic C-H groups and glycoside oxygen atoms). The outer layer is hydrophilic, resulting in good solubility of cyclodextrins in water. This arrangement of the molecule allows cyclodextrins to form stable inclusion complexes (host-guest complexes) with many ions and organic compounds in a molar ratio of 1:1 without the need to form covalent bonds [2]. This process, often called microencapsulation, can take place in both an aqueous solution and a solid phase. This enables the formation of complexes diversified in composition with a wide range of applications in many areas, including in the food [2,3,4], pharmaceutical [5,6], cosmetic [7,8] and textile [9] industries. Numerous examples of the use of cyclodextrins in production [9], environmental protection [10,11], and biocatalysis [12] have also been described. Data from the literature also show that these compounds can be used successfully to separate isomers, enantiomers, or compounds with different functional groups [13]. Due to their properties and the presence of a chiral cavity, cyclodextrins dominate the analysis of optical isomers and become called chiral stationary phases [14,15].

Commercially available cyclodextrins are sold as complexes with water. However, these complexes are unstable and water molecules can easily be replaced by guest molecules. There are also many modified cyclodextrin derivatives. Typically, they are produced by the esterification or etherification of the primary and secondary cyclodextrin hydroxyl groups. These derivatives differ from the starting compounds in the size of the hydrophobic hole. Their complexes are often characterized by better solubility and lower sensitivity to light and oxygen [3]. In the food industry, cyclodextrin inclusion complexes are used on a large scale to protect and stabilize substances sensitive to moisture, light, or oxygen, to modify the physicochemical properties of the sample, eg solubility, volatility, to mask undesirable color, smell and taste of selected products [16] or to bind volatile and highly toxic substances, which often leads to a significant extension of storage time [2]. The number of free and complexed guest molecules in the aqueous solution of cyclodextrins depends on several factors, the most important of which are the complexation constant, the temperature, and the concentration of both components. In low-temperature and high-concentration solutions, the equilibrium of this process is shifted toward the formation of inclusion complexes (and crystallization).

The most common hydroxybenzoic acids are gallic, p-hydroxybenzoic, protocatechic, vanillic, and syringic acids, and hydroxycinnamic acids: coffee, ferulic, p-coumaric, and sinapic.

In plants, phenolic acids are generally bound in the form of esters and glycosides, which are part of lignins and hydrolyzing tannins. Some of the hydroxycinnamic acids are found in ester combinations with carboxylic acids or glucose. They occur in ester combinations with the following acids: malonic, tartaric, α-hydroxy-hydrocavic, hydroxycinnamic, tartronic (HOOC-CHOH-COOH, as p-coumaroyl, feruloyl, and caffeoyl-tartronic acid), shikimic, galactaric, glucaric acid (as caffeoylglucaric acid), glucose (as feruloylgluconic acid, the main isomer of which is 2-o-feruloylgluconic acid), 4-methoxyaldaric acid (as 2-o-feruloyl-4-methoxyaldaric acid). In an acidic environment, when heated, these compounds can undergo hydrolysis, which breaks ester and glycosidic bonds, which in turn leads to an increase in the number of free phenolic acids.

On the contrary, hydroxybenzoic acids are mostly present as glycosides. Other combinations of phenolic acids, e.g., flavonoids, fatty acids, sterols, and cell wall polymers, have also been defined in plant tissues. Phenolic acids can also be components of anthocyanins or flavones.

Hydroxycinnamic acid derivatives are very widespread in plants. It has been shown that they can act as an antioxidant by removing free radicals (which is related to their ability to donate a proton and, as a result of this reaction, create a stable and not very reactive phenoxy radical). They mainly form esters with organic acids and glycosides [17]. As indicated in the literature, esterification of caffeic acid with a simple carbohydrate reduces its antioxidant properties [18]. Chen and Ho proved that chlorogenic acid showed lower antioxidant activity compared to caffeic acid contained in corn oil [19]. Caffeic acid removed alkoxy radicals (which were the result of the decomposition of methyl esters of linoleic acid in sunflower oil) much better than chlorogenic acid. Additionally, caffeic acid is more effective in preventing lipid oxidation in fish muscles than chlorogenic acid [20]. On the other hand, Sroka and Lisowski suggest that the addition of quinonic acid to caffeic acid increases antioxidant activity, but also reduces the scavenging capacity of DPPH radicals compared to caffeic acid alone [21]. Cheng, Dai, and Zhou demonstrated that both acids prevent LDL peroxidation in vitro equally [22]. Marinova’s research showed that the antioxidant activity of both acids during the oxidation of triacylglycerols in sunflower oil strongly depends on the concentration of both antioxidants. The use of both acids at a concentration of 2.8·10^−4^ M allowed the evidence that both have similar antioxidant activity. However, the use of both acids at concentrations above 2.8·10^−4^ M revealed that caffeic acid is a much more effective reaction inhibitor than chlorogenic acid. Specific interactions may explain this phenomenon between acid and hydroperoxides, as well as the possibility of participation of free radical derivatives of chlorogenic acid in more than one propagation reaction, when free radical derivatives of caffeic acid participate in only one [23].

## 2. Results and Discussion

The basic values of density, viscosity and dielectric permittivity for the solvent used are summarized in Appendix A. With the help of these data, it was possible to determine the values of the tested molar concentrations. The molar conductivities values of the tested phenolic acid derivatives from *β*-cyclodextrin and 2-HP-*β*-cyclodextrin are presented in Appendix A. The molar concentration values presented in the tables were calculated from the dependencies presented in the article [24]. In the discussed study *β*-cyclodextrin and 2-HP-*β*-cyclodextrin with the investigated anions of phenolic acid form complexes in a 1:1 ratio according to the following relationship:$$L+Na^{+}↔LNa^{+}$$

The molar conductivity of the tested solution is described by the following relationship:(1)$$\Lambda_{\mathrm{obs}}=\alpha \cdot \Lambda_{\mathrm{NaDod}}+\left(1-\alpha \right)\cdot \Lambda_{\mathrm{CDNaDod}}$$
where:(2)$$\alpha =\frac{\left[{\mathrm{Dod}}^{-}\right]}{\left[{\mathrm{Na}}^{+}\right]}$$
(3)$$1-\alpha =\frac{\left[{\mathrm{DodCD}}^{-}\right]}{\left[{\mathrm{Na}}^{+}\right]}$$
(4)$$C_{\mathrm{DodNa}}=\left[{\mathrm{Na}}^{+}\right]=\left[{\mathrm{Dod}}^{-}\right]+\left[{\mathrm{DodCD}}^{-}\right]$$
(5)$$C_{\mathrm{CD}}=\left[\mathrm{CD}\right]+\left[{\mathrm{DodCD}}^{-}\right]$$

The constant of complex ion (DodCD^−^) formation is given by the equation:(6)$$K_{f}=\frac{\left[{\mathrm{DodCD}}^{-}\right]}{\left[{\mathrm{Dod}}^{-}\right]\cdot \left[\mathrm{CD}\right]}$$

Taking into account Equations (4)–(6) we get:(7)$$K_{f}\cdot {\left[\mathrm{CD}\right]}^{2}+\left[K_{f}\cdot \left(C_{\mathrm{DodNa}}-C_{\mathrm{CD}}\right)+1\right]\cdot \left[\mathrm{CD}\right]-C_{\mathrm{CD}}=0$$

Thanks to the new equations proposed in the paper [24], which allow assessing the values of the formation constants and theoretical conductivity, we can analyze the correctness of these equations from the point of view of the size of cyclodexrins and complexed ions. The obtained results of the conductometric measurements allow us to determine the equilibrium constants of this process using the calculation procedures described in detail in our previous work [24]:(8)$$K_{f}=\frac{\left[LNa^{+}\right]}{\left[L\right]\left[Na^{+}\right]}=\frac{\left(1-\alpha \right)C_{M^{+}}}{\alpha C_{M^{+}}\left[L\right]}=\frac{\left(1-\alpha \right)}{\alpha \left[C_{L}-\left(1-\alpha \right)C_{M^{+}}\right]}$$

Thus, the complexation constant can be represented by the following relationship.
(9)$$\Lambda =\left[K_{f}(c_{DodNa}-C_{Cd}-1+\sqrt{K_{f}^{2}{\left(C_{Cd}-C_{DodNa}\right)}^{2}+2K_{f}\left(C_{DodNa}+C_{Cd}\right)+1}\right]\cdot \left[\frac{\Lambda_{NaDod}-\Lambda_{NaDodCd}}{2K_{f}C_{DodNa}}\right]+\Lambda_{NaDodCD}$$

On the other hand, the values for *Λ_NaDod_* and *Λ_NaDodCD_* show dependencies:(10)$$\Lambda_{NaDod}=\Lambda_{0NaDod}-Sc_{NaDod}{\;}^{\frac{1}{2}}+Ec_{NaDod}lnc_{NaDod}+J_{1}c_{S}+J_{2}c_{NaDod}.^{\frac{3}{2}}$$
(11)$$\Lambda_{NaDodCD}=\Lambda_{0NaDodCD}-Sc_{NaDod}{\;}^{\frac{1}{2}}+Ec_{NaDod}lnc_{NaDod}+J_{1}c_{S}+J_{2}c_{NaDod}.^{\frac{3}{2}}$$

The functions discussed have already been described and analyzed many times in the literature [24,25,26].

The molar conductivity as a function of concentration was analyzed in Appendix A. These values increase as a function of concentration and as a function of temperature. The increase in molar conductivity is fully understandable based on the theory of conductivity because the mobility of the ions increases at higher temperatures. The interactions between ions and solvent molecules become weaker and weaker, which causes an increase in the values of the molar conductivity. As can be seen, an increase in the concentration of the tested ligand (i.e., *β*-cyclodextrin and 2-HP-*β*-cyclodextrin), we observe a decrease in the conductivity value. This decrease is most likely due to the formation of inclusion complexes of the tested phenolic acid anions with the tested ligands. The *K_f_* values and the limiting molar conductivities for both studied cyclodextrins are summarized in Table 1, Table 2, Table 3 and Table 4.

As can be seen, the values of *K_f_* increase with increasing molar mass of the tested anion (cinnamon > coumarin > coffee > ferulic) for all studied cyclodextrins, which can be seen in Figure 1.

It should be emphasized that these values decrease with an increase in the molar mass of the tested cyclodextrin. This indicates that the stability of the investigated complexes increases with the decrease in the cyclodextrin molar mass. This relationship is also valid if we take into account the *K_f_* values previously obtained for α-cyclodextrins [24]. Thus, the analyzed values of the *K_f_* formation constants as a function of the cyclodextrin molar mass satisfy the following relationship. *K_f_* (*α*-cyclodexrin) > *K_f_* (*β*-cyclodextrin) > *K_f_* (2-hydroxypropyl-*β*-cyclodextrin). It should also be noted that with lowering the temperature, the stability of the analyzed complexes increases. This is evident for all tested cyclodextrins. This is most likely due to the reduction in the acids mobility of the ions tested by lowering the temperature. At high temperatures, the ion-solvent interactions are weak and the mobility of ions increases, which is not conducive to the formation of stable inclusion complexes. On the other hand, at low temperatures, the mobility of the discussed anions is relatively low and, most likely, their matching to the gap in the cone of the tested cyclodextrin is more precise, which favors the formation of the complex.

The size of the cone of the tested cyclodextrins is also very important for the process discussed. As the interior of cyclodextrin increases, the complexing power decreases. The strongest complexes are noticeable for α-cyclodextrin [24], they are weaker for *β*-cyclodextrin, and for 2-HP-*β*-CD they are the weakest. This is due to the fact that they have substituents in their structure in addition to the hydrophobic cone, which may weaken the power of the complexes formed. In the case of 2-HP-*β*-CD, 5 to 6 2-hydroxypropyl groups are attached to the lower cone. This may make the interior of the cyclodextrin less hydrophobic, and also it is due to the fact that the anions of the phenolic acid derivatives that attach to the analyzed cyclodextrin cannot fully fill its interior. This probably makes the complex formation process less spontaneous for both *β*-cyclodextrins. The fact that in the case of complexes of both *β*-cyclodextrins the values of the formation constants decrease can be explained by an increase in the kinetic energy of the anions and cyclodextrin molecules. However, this does not mean that the spontaneity of the formation process decreases with increasing temperature. This problem can be solved by analysing the free formation enthalpies, which are presented later in the paper.

The molar conductivities of salts composed of sodium cations and complexed phenolic anions increase monotonically with increasing temperature. Data contained in Table 1, Table 2, Table 3 and Table 4 and the relations presented in Figure 2. 

Show that the lowest values of the formation constants of the inclusion complex and an increase in their solubility were achieved using 2HP-*β*-CD in the complexation reaction of the phenolic acid derivatives. 

These effects are more noticeable with the use of 2HP-*β*-CD than with unsubstituted *β*-CD studies. This phenomenon may be explained by the fact that the substituents extend the cyclodextrin gap, resulting in a more efficient complexation of phenolic anions that have a fairly flat linear structure. The above data also show that cyclodextrins with larger gap sizes, such as *β*-cyclodextrin and its derivatives, can effectively complex the tested phenolic derivatives, thus increasing their solubility in aqueous solutions. This structure suggests the use of cyclodextrins, which dissolve well in water [27,28,29,30,31], and at the same time can increase the solubility of soluble substances by including hydrophobic groups inside the cyclodextrin gap [27,28,29,30,31].

Temperature dependences of the complex formation constant were used to determine the free enthalpy of the complex formation.
(12)$$∆G\left(T\right)=-RTlnK_{f}\left(T\right)$$

The above relationship can be presented as follows.
(13)$$∆G\left(T\right)=A+BT+CT^{2}$$

The enthalpy values are presented as the first derivative of the free enthalpy after temperature at constant pressure.
(14)$$∆S^{0}=-{\left(\frac{\partial ∆G^{o}}{\partial T}\right)}_{p}=-B-2CT$$

The enthalpy is calculated from the following relationship:(15)$$∆H^{0}=∆G^{0}+T∆S^{0}=A-CT^{2}$$

Using the parameters obtained for the values a and b, the values of Δ*G*^0^, Δ*H*^0^ and Δ*S*^0^ were determined for the formation of inclusion complexes between the phenolic acid derivatives and *β*-CD and 2HP-*β*-CD. The values of the thermodynamic functions are summarized in Table 5, Table 6, Table 7 and Table 8.

As can be seen, for both types of inclusion complexes (with *β*-CD and 2HP-*β*-CD) the values of ∆*H*^0^, ∆G^0^ and ∆*S*^0^ are <0, which means that the formation of inclusion complexes is exothermic and runs spontaneously. Comparing the values in Table 5, Table 6, Table 7 and Table 8, it can be seen that in the case of complexes with modified cyclodextrin, the complexation reactions are less spontaneous (the values of the thermodynamic functions are less negative). 

As can be seen in Figure 3, Figure 4 and Figure 5, as the temperature increases, the spontaneity of the inclusion complex formation process increases contrary to what is shown in the analysis of changes in the formation constants. The higher energy of the anions favors the penetration of the cyclodextrin interior, and the molecules are located inside it more intensively. However, as can be seen, the energy gain in the complex formation process is clear for both cyclodextrins and much greater for unsubstituted *β*-cyclodextrin. The entropy of the complex formation process decreases with increasing temperature and increasing the molar mass of the tested cyclodextrin and the molar mass of the tested anions. 

These values are higher for *β*-cyclodextrin. This probably means that the total change in the dehydration effects of the hydrophobic part of the phenolic anion before its location inside cyclodextrin leads to an increase in the entropy. The larger and deeper location of the anions studied in unsubstituted cyclodextrin is associated with a greater level of hydrophobic dehydration and thus with a slightly greater increase in entropy. It is worth bearing in mind at this point that hydrophobic hydration is associated with a decrease in entropy. 

The loss of hydrophobic hydration may result in a change in entropy. As can be seen, the formation of the studied complexes from the point of view of their spontaneity is determined by changes in the free enthalpy (into the results of energy gain associated with intermolecular interactions). Thermodynamic analysis of the studied complexing processes shows that in order for complexing to occur, an appropriate change in the free enthalpy must occur.

There are a number of interactions that allow the equilibrium to be shifted towards complex formation. However, the main driving force is to replace the polar-nonpolar interactions between the water and CD molecules, and the water and drug molecules, with the nonpolar-nonpolar interactions between the drug molecule and cyclodextrin.

A review of data from the literature shows that there are many research techniques (spectroscopic, electroanalytical, and separation techniques) allowing to assess the stability of cyclodextrin inclusion complexes with various types of compounds that have potential use as drugs [32]. As the authors of the works presented in the literature emphasize, none of the research techniques used separately allows for a complete structural assessment of the forming complexes [32,33,34]. They only allow one to determine the stoichiometry of the forming complexes. In the presented work, we proposed stoichiometry 1:1 for the studied complexes. Polewski et al. [33] proposed a similar stoichiometry for chlorogenic and cinnamic acids with beta-cyclodextrin based on a spectroscopic, thermodynamic and molecular modeling study. Similar conclusions can be drawn from the work of Plasson et al. [34]. In their work, affinity capillary electrophoresis was used for the simultaneous measurement of the p*K*a values and of the binding constants relative to the encapsulation of naturally occurring phenolic acids (rosmarinic and caffeic acids) with cyclodextrin. It should be emphasized that in the cited articles [33,34], all thermodynamic functions describing complexation processes (regardless of the research technique used) are negative, as in the case of the results presented by us. This indicates the same mechanism for the formation of inclusion complexes. The presented works (especially the work [32]) show how difficult it is to determine the exact stoichiometry and structure of the forming complexes.

The results of conductometric measurements presented in this paper only allow us to propose the stoichiometry of the forming inclusion complexes. Conductometry is one of the most accurate research techniques that allows for conducting this type of research. The innovative computational methods introduced by us in the previous work [24] and used in this work allow an accurate assessment of the complexation process taking place in the tested systems. Unfortunately, similar to other research methods used in the literature [32], it does not allow a full structural assessment of the inclusion complexes based on the results. Therefore, in order to supplement this information, we plan future spectroscopic and molecular modeling studies in the mixtures containing both α- and β-cyclodextrins.

## 3. Materials and Methods

### 3.1. Materials

High purity of *trans*-4-hydroxycinnamic acid (*trans-p*-coumaric acid), *trans*-3,4-dihydroxycinnamic acid (*trans*-caffeic acid), *trans*-4-hydroxy-3-methoxycinnamic acid (*trans*-ferulic acid), *trans*-3-phenylacrylic acid (*trans*-cinnamonic acid) and β-cyclodextrin and 2-HP-βcyclodextrin were used. The water used for the measurements was distilled twice and then passed through an ion exchanger to obtain the best purity (Behr Laboor-Technik, Düsseldorf, Germany). The water was distilled twice (specific conductivity ~ 10^−6^ S‧cm^−1^).

All the information on their purity and suppliers is presented in Table 9.

### 3.2. Characterization Methods

Sodium salts of the tested phenolic acids were used for the measurements. The salts studied were obtained by mixing the appropriate amounts of acid and aqueous sodium hydroxide solution in a stoichiometric ratio of 1:1. The mixture was then heated and stirred to dissolve the acid and evaporate the solvent. All analyzed sodium salts of acids were recrystallized twice from aqueous ethanol solutions. The salts obtained were washed with acetone and dried under reduced pressure in a Büchi glass oven B-580 at T = 373.15 K until constant weight.

The exact procedure for their preparation is described in the work [24,35,36]. All sodium salts of the phenolic acids prior to measurement were dried and placed in a desiccator to avoid contact with ambient moisture.

Conductivity measurements were made on the highest-class RLC Wayne-Kerr 6430B conductometer with an uncertainty of 0.02%. The measuring cell was a three-electrode cell made of Pyrex glass containing no sodium in its composition (which would interfere with the measurement). The electrodes in this cell were made of platinum. The measuring cell in the tested temperature range 283.15–313.15 K was calibrated using KCl as the standard substance. All measurements were made at different frequencies *v* = 0.2, 0.5, 1.0, 1.5, 2.0, 3.0, 5.0, 10.0 and 20.0 kHz to obtain the most accurate values of the molar conductivity.

For all measurements, a BU 20F calibration thermostat (Lauda, Germany) with stability better than 0.005 K was used and the temperature was checked using an Amarell 3000TH AD thermometer (Amarell, Germany). The thermostat was connected to a DLK 25 flow cooler (Lauda, Germany). The entire measurement procedure is described in detail in [37,38,39]. All measured conductivity values, *λ = 1/R_∞_,* were the results of extrapolation of cell resistance, *R_∞_(ν)*, to infinite frequency *R_∞_ = lim_ν→∞_R(ν)* using the empirical function *R*(*ν*) = *R_∞_ + A/ν* (where parameter *A* is specific for the cell). Taking into account the sources of errors (calibration, sample purity, measurements), the estimated uncertainty of the measured conductivity values was estimated at ±0.05%.

## 4. Conclusions

The paper presents conductometric measurements of phenolic acid derivatives with *β* and 2HP-*β*-CD cyclodextrin in aqueous solutions over a wide temperature range. The values of thermodynamic functions determined on the basis of experimental measurements (*ΔH*^0^ < 0, *ΔS*^0^ < 0, *ΔG*^0^ < 0) indicate the spontaneity of the inclusion of fungicides in the hydrophobic interior of the macromolecule of both cyclodextrins. The stoichiometry of the inclusion complexes of the investigated phenolic derivatives with both cyclodextrins is type 1:1. Conductometric measurements allowed us to estimate the stability of the formed complexes. It was found that the power of these complexes decreases with the size of the cyclodextrins and also with increasing temperature. This effect is most likely related to an increase in the mobility of ions of the tested phenolic acids, which significantly effects interactions of the type: the anion of studied phenolic acid, cyclodextrin.

## Figures and Tables

**Figure 1 molecules-28-00292-f001:**
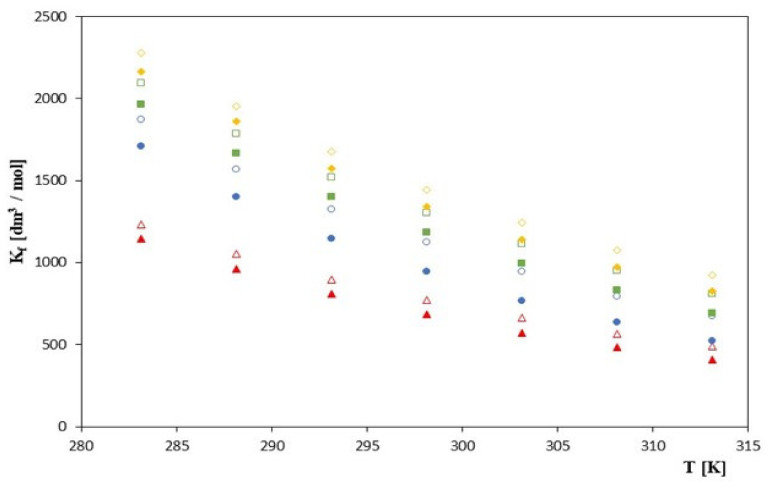
The plots of dependence of the *K_f_* [dm^3^/mol] formation constant in the *T* [K] function for *β*-cyclodextrin with all studied salts.: 
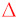
-*trans*-cinnamic acid; 
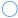
-*trans*–p-coumaric acid; 
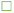
-*trans*-caffeic acid; 
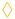
-*trans*-ferulic acid and 2-HP-*β*-cyclodextrin with the studied salts.: 
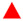
-*trans*-cinnamic acid; 
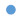
-*trans*–p-coumaric acid; 

-*trans*-caffeic acid; 
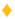
-*trans*-ferulic acid.

**Figure 2 molecules-28-00292-f002:**
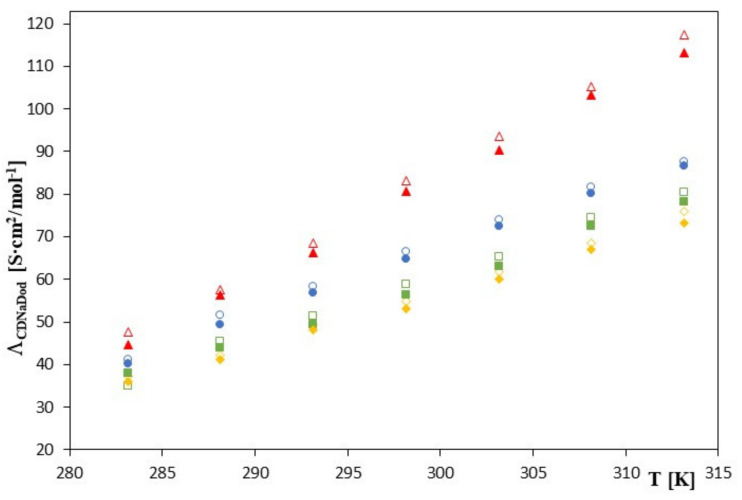
The plots of the dependence of the theoretical conductivity Λ_CDNaDod_ [S∙cm^2^/mol^−1^] in the function *T* [K] for *β*-cyclodextrin with all studied salts.: 
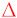
-*trans*-cinnamic acid; 
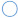
-*trans*–p-coumaric acid; 
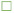
-*trans*-caffeic acid; 
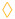
-*trans*-ferulic acid and 2-HP-*β*-cyclodextrin with the studied salts.: 
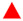
-*trans*-cinnamic acid; 
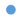
-*trans*–p-coumaric acid; 

-*trans*-caffeic acid; 
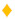
-*trans*-ferulic acid.

**Figure 3 molecules-28-00292-f003:**
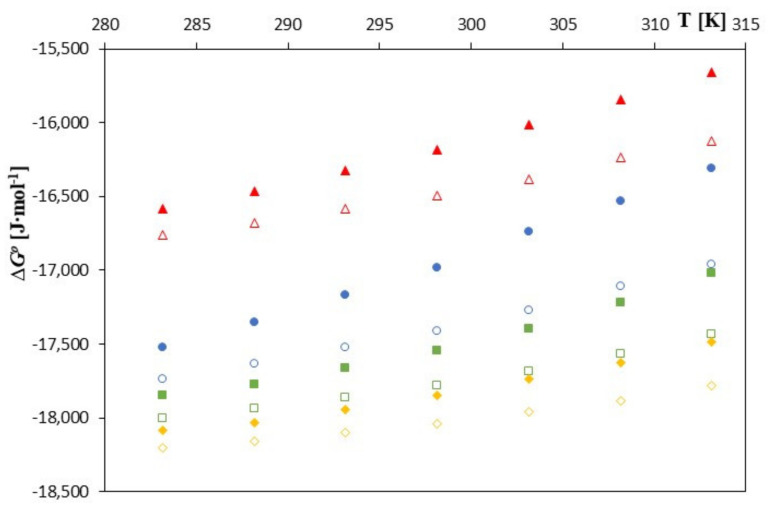
The plots of dependence ∆*G*^0^ [J∙mol^−1^] in the function *T* [K] for *β*-cyclodextrin with all studied salts.: 
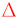
-*trans*-cinnamic acid; 
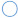
-*trans*–p-coumaric acid; 
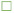
-*trans*-caffeic acid; 
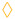
-*trans*-ferulic acid and 2-HP-*β*-cyclodextrin with the studied salts.: 
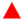
-*trans*-cinnamic acid; 
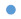
-*trans* –p-coumaric acid; 

-*trans*-caffeic acid; 
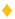
-*trans*-ferulic acid.

**Figure 4 molecules-28-00292-f004:**
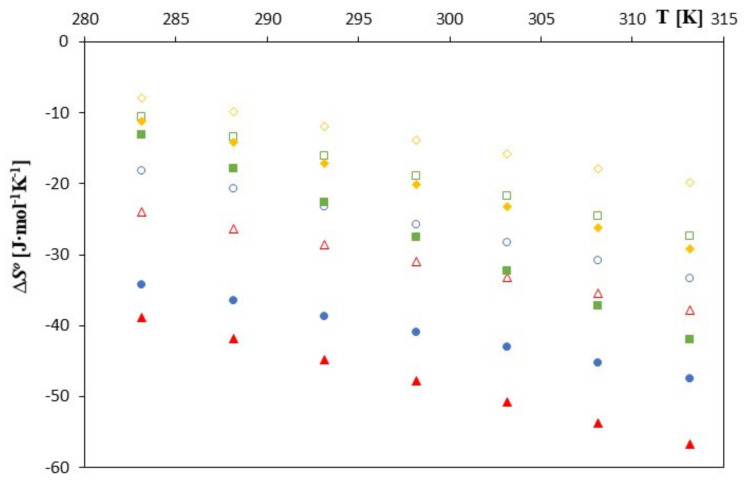
The plots of the dependence ∆*S*^0^ [J∙mol^−1^] in the function *T* [K] for *β*-cyclodextrin with all studied salts.: 
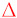
-*trans*-cinnamic acid; 
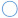
-*trans*–p-coumaric acid; 
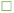
-*trans*-caffeic acid; 
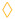
-*trans*-ferulic acid and 2-HP-*β*-cyclodextrin with the studied salts.: 
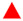
-*trans*-cinnamic acid; 
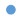
-*trans*–p-coumaric acid; 

-*trans*-caffeic acid; 
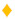
-*trans*-ferulic acid.

**Figure 5 molecules-28-00292-f005:**
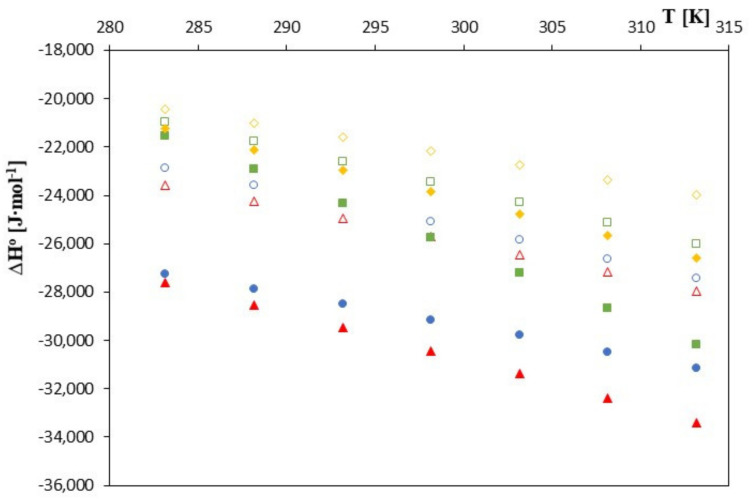
The plots of dependence ∆H^0^ [J∙mol^−1^] in the function *T* [K] for *β*-cyclodextrin with all studied salts.: 
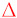
-*trans*-cinnamic acid; 
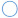
-*trans*–p-coumaric acid; 
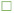
-*trans*-caffeic acid; 
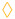
-*trans*-ferulic acid and 2-HP-*β*-cyclodextrin with the studied salts.: 
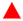
-*trans*-cinnamic acid; 
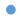
-*trans*–p-coumaric acid; 

-*trans*-caffeic acid; 
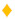
-*trans*-ferulic acid.

**Table 1 molecules-28-00292-t001:** The value of constant formation *K_f_* [dm^3^/mol], theoretical conductivity Λ*_CDNaDod_* [S∙cm^2^/mol^−1^] for *β*-cyclodextrin and 2-HP-*β*-cyclodextrin with the salt of *trans*-cinnamic acid.

β-Cyclodextrin					2-HP-β-Cyclodextrin			
T [K]	*K_f_* [dm^3^/mol]	ln*K_f_* [dm^3^/mol]	Λ*_CDNaDod_* [S∙cm^2^/mol^−1^]	σ(Λ)	*K_f_* [dm^3^/mol]	ln*K_f_* [dm^3^/mol]	Λ*_CDNaDod_* [S∙cm^2^/mol^−1^]	σ(Λ)
283.15	1235 ± 8	7.1188	47.56 ± 0.02	0.01	1145 ± 8	7.0432	44.75 ± 0.01	0.01
288.15	1055 ± 4	6.9613	57.55 ± 0.01	0.01	965 ± 6	6.8721	56.23 ± 0.01	0.02
293.15	900 ± 2	6.8024	68.52 ± 0.01	0.02	810 ± 4	6.6970	66.32 ± 0.02	0.02
298.15	775 ± 2	6.6529	83.235 ± 0.02	0.02	685 ± 3	6.5294	80.75 ± 0.01	0.01
303.15	665 ± 3	6.4998	93.62 ± 0.01	0.01	575 ± 3	6.3544	90.45 ± 0.01	0.01
308.15	565 ± 2	6.3368	105.24 ± 0.01	0.01	485 ± 2	6.1841	103.23 ± 0.01	0.01
313.15	489 ± 0.9	6.1924	117.32 ± 0.01	0.01	409 ± 1	6.0137	113.15 ± 0.01	0.02

**Table 2 molecules-28-00292-t002:** The value of constant formation *K_f_* [dm^3^/mol], theoretical conductivity Λ*_CDNaDod_* [S∙cm^2^/mol^−1^] for *β*-cyclodextrin and 2-HP-*β*-cyclodextrin with the salt of *trans*–*p*-coumaric acid.

β-Cyclodextrin					2-HP-β-Cyclodextrin			
T [K]	*K_f_* [dm^3^/mol]	ln*K_f_* [dm^3^/mol]	Λ*_CDNaDod_* [S∙cm^2^/mol^−1^]	σ(Λ)	*K_f_* [dm^3^/mol]	ln*K_f_* [dm^3^/mol]	Λ*_CDNaDod_* [S∙cm^2^/mol^−1^]	σ(Λ)
283.15	1870 ± 7	7.5337	41.24 ± 0.01	0.01	1710 ± 6	7.4442	40.11 ± 0.01	0.02
288.15	1570 ± 4	7.3588	51.73 ± 0.01	0.01	1400 ± 5	7.2442	49.52 ± 0.02	0.02
293.15	1324 ± 3	7.1884	58.23 ± 0.02	0.02	1144 ± 4	7.0423	56.85 ± 0.01	0.02
298.15	1124 ± 3	7.0246	66.56 ± 0.01	0.02	944 ± 2	6.8501	64.78 ± 0.02	0.01
303.15	945 ± 2	6.8512	73.87 ± 0.02	0.01	765 ± 1	6.6399	72.54 ± 0.01	0.01
308.15	795 ± 1	6.6783	81.56 ± 0.01	0.01	635 ± 0.9	6.4536	80.12 ± 0.01	0.01
313.15	675 ± 0.8	6.5147	87.52 ±0.01	0.02	525 ± 0.9	6.2634	86.65 ± 0.02	0.02

**Table 3 molecules-28-00292-t003:** The value of constant formation *K_f_* [dm^3^/mol], theoretical conductivity Λ*_CDNaDod_* [S∙cm^2^/mol^−1^] for *β*-cyclodextrin and 2-HP-*β*-cyclodextrin with the salt of *trans*-caffeic acid.

β-Cyclodextrin					2-HP-β-Cyclodextrin			
T [K]	*K_f_* [dm^3^/mol]	ln*K_f_* [dm^3^/mol]	Λ*_CDNaDod_* [S∙cm^2^/mol^−1^]	σ(Λ)	*K_f_* [dm^3^/mol]	ln*K_f_* [dm^3^/mol]	Λ*_CDNaDod_* [S∙cm^2^/mol^−1^]	σ(Λ)
283.15	2092 ± 6	7.6459	35.12 ± 0.01	0.01	1962 ± 6	7.5817	38.05 ± 0.01	0.01
288.15	1785 ± 4	7.4872	45.55 ± 0.01	0.01	1665 ± 5	7.4176	43.89 ± 0.01	0.02
293.15	1522 ± 4	7.3278	51.42 ± 0.02	0.02	1402 ± 3	7.2457	49.52 ± 0.01	0.02
298.15	1305 ± 2	7.1740	58.74 ± 0.01	0.02	1185 ± 2	7.0775	56.32 ± 0.02	0.01
303.15	1115 ± 1	7.0166	65.32 ± 0.01	0.01	995 ± 1	6.9027	63.12 ± 0.02	0.01
308.15	950 ± 0.9	6.8565	74.59 ± 0.01	0.01	830 ± 1	6.7214	72.56 ± 0.01	0.01
313.15	810 ± 0.8	6.6970	80.45 ± 0.01	0.01	690 ± 0.9	6.5367	78.15 ± 0.01	0.01

**Table 4 molecules-28-00292-t004:** The value of constant formation *K_f_* [dm^3^/mol], theoretical conductivity Λ*_CDNaDod_* [S∙cm^2^/mol^−1^] for *β*-cyclodextrin and 2-HP-*β*-cyclodextrin with the salt of *trans*-ferulic acid.

β-Cyclodextrin					2-HP-β-Cyclodextrin			
T [K]	*K_f_* [dm^3^/mol]	ln*K_f_* [dm^3^/mol]	Λ*_CDNaDod_* [S∙cm^2^/mol^−1^]	σ(Λ)	*K_f_* [dm^3^/mol]	ln*K_f_* [dm^3^/mol]	Λ*_CDNaDod_* [S∙cm^2^/mol^−1^]	σ(Λ)
283.15	2280 ± 7	7.7319	37.28 ± 0.01	0.02	2165 ± 5	7.6802	36.11 ± 0.01	0.02
288.15	1955 ± 6	7.5781	42.31 ± 0.01	0.01	1860 ± 5	7.5283	41.10 ± 0.01	0.02
293.15	1680 ± 5	7.4265	49.47 ± 0.02	0.02	1575 ± 4	7.3620	48.11 ± 0.01	0.02
298.15	1446 ± 3	7.2766	54.95 ± 0.01	0.02	1341 ± 4	7.2012	53.05 ± 0.01	0.01
303.15	1244 ± 2	7.1261	61.73 ± 0.02	0.01	1139 ± 3	7.0379	60.1 ± 0.01	0.01
308.15	1076 ± 2	6.9810	68.41 ± 0.01	0.01	971 ± 2	6.8783	67.14 ± 0.01	0.01
313.15	925 ± 1	6.8298	75.95 ± 0.01	0.02	825 ± 1	6.7154	73.12 ± 0.02	0.01

**Table 5 molecules-28-00292-t005:** The values of thermodynamic functions ∆*G*^0^, ∆*S*^0^, ∆*H*^0^ [J∙mol^−1^] for *β*-cyclodextrin and 2-HP-*β*-cyclodextrin with the salt of *trans*-cinnamic acid.

β-Cyclodextrin				2-HP-β-Cyclodextrin		
T [K]	∆*G*^0^ [J∙mol^−1^]	∆*S*^0^ [J∙mol^−1^]	∆*H*^0^ [J∙mol^−1^]	∆*G*^0^ [J∙mol^−1^]	∆*S*^0^ [J∙mol^−1^‧K^−1^]	∆*H*^0^ [J∙mol^−1^]
283.15	−16,758	−24.036	−23,565	−16,580	−38.888	−27,592
288.15	−16,677	−26.319	−24,261	−16,463	−41.844	−28,521
293.15	−16,579	−28.602	−24,964	−16,322	−44.800	−29,456
298.15	−16,491	−30.885	−25,700	−16,185	−47.756	−30,424
303.15	−16,382	−33.168	−26,437	−16,016	−50.712	−31,389
308.15	−16,235	−35.451	−27,159	−15,844	−53.668	−32,382
313.15	−16,122	−37.734	−27,939	−15,657	−56.624	−33,389

**Table 6 molecules-28-00292-t006:** The values of thermodynamic functions ∆*G*^0^, ∆*S*^0^, ∆*H*^0^ [J∙mol^−1^] for *β*-cyclodextrin and 2-HP-*β*-cyclodextrin with the salt of *trans* –*p*-coumaric acid.

β-Cyclodextrin				2-HP-β-Cyclodextrin		
T [K]	∆*G*^0^ [J∙mol^−1^]	∆*S*^0^ [J∙mol^−1^]	∆H^0^ [J∙mol^−1^]	∆*G*^0^ [J∙mol^−1^]	∆*S*^0^ [J∙mol^−1^‧K^−1^]	∆*H*^0^ [J∙mol^−1^]
283.15	−17,735	−18.180	−22,883	−17,525	−34.296	−27,235
288.15	−17,629	−20.714	−23,598	−17,355	−36.486	−27,868
293.15	−17,520	−23.248	−24,335	−17,164	−38.676	−28,502
298.15	−17,413	−25.782	−25,100	−16,980	−40.866	−29,164
303.15	−17,268	−28.316	−25,852	−16,735	−43.056	−29,787
308.15	−17,110	−30.850	−26,616	−16,534	−45.246	−30,476
313.15	−16,961	−33.384	−27,416	−16,307	−47.436	−31,161

**Table 7 molecules-28-00292-t007:** The values of thermodynamic functions ∆*G*^0^, ∆*S*^0^, ∆*H*^0^ [J∙mol^−1^] for *β*-cyclodextrin and 2-HP-*β*-cyclodextrin with the salt of *trans*-caffeic acid.

β-Cyclodextrin				2-HP-β-Cyclodextrin		
T [K]	∆*G*^0^ [J∙mol^−1^]	∆*S*^0^ [J∙mol^−1^]	∆*H*^0^ [J∙mol^−1^]	∆*G*^0^ [J∙mol^−1^]	∆*S*^0^ [J∙mol^−1^‧K^−1^]	∆*H*^0^ [J∙mol^−1^]
283.15	−17,999	−10.534	−20,982	−17,848	−13.040	−21,540
288.15	−17,937	−13.343	−21,782	−17,770	−17.865	−22,918
293.15	−17,860	−16.152	−22,594	−17,660	−22.690	−24,311
298.15	−17,783	−18.961	−23,436	−17,544	−27.515	−25,747
303.15	−17,685	−21.770	−24,284	−17,398	−32.340	−27,201
308.15	−17,566	−24.579	−25,140	−17,220	−37.165	−28,672
313.15	−17,436	−27.388	−26,012	−17,018	−41.990	−30,168

**Table 8 molecules-28-00292-t008:** The values of thermodynamic functions ∆*G*^0^, ∆*S*^0^, ∆*H*^0^ [J∙mol^−1^] for *β*-cyclodextrin and 2-HP-*β*-cyclodextrin with the salt of *trans*-ferulic acid.

β-Cyclodextrin				2-HP-β-Cyclodextrin		
T [K]	∆*G*^0^ [J∙mol^−1^]	∆*S*^0^ [J∙mol^−1^]	∆*H*^0^ [J∙mol^−1^]	∆*G*^0^ [J∙mol^−1^]	∆*S*^0^ [J∙mol^−1^‧K^−1^]	∆*H*^0^ [J∙mol^−1^]
283.15	−18,202	−7.9240	−20,445	−18,080	−11.204	−21,252
288.15	−18,155	−9.8990	−21,007	−18,035	−14.191	−22,125
293.15	−18,101	−11.874	−21,581	−17,943	−17.178	−22,979
298.15	−18,037	−13.849	−22,166	−17,850	−20.165	−23,863
303.15	−17,960	−15.824	−22,758	−17,738.	−23.152	−24,757
308.15	−17,885	−17.799	−23,370	−17,622	−26.139	−25,677
313.15	−17,782	−19.774	−23,974	−17,484	−29.126	−26,604

**Table 9 molecules-28-00292-t009:** Specification of chemical samples.

Chemical Name	Chemical Formula	Chemical Formula	Source	CAS No	Mass Fraction Purity
Trans-ferulic acid	C_10_H_10_O_4_	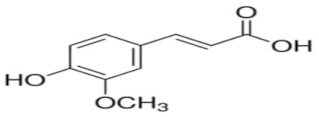	TCI *	537-98-4	≥0.998
Trans-caffeic acid	C_9_H_8_O_4_	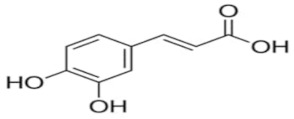	TCI *	331-39-5	≥0.998
Trans –p-coumaric acid	C_9_H_8_O_3_	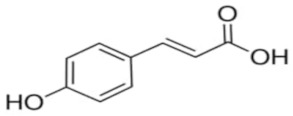	TCI *	501-98-4	≥0.998
Trans-cinnamic acid	C_9_H_8_O_2_	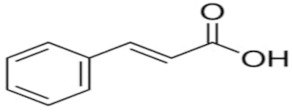	TCI *	140-10-3	≥0.998
β -cyclodextrin	C_42_H_70_O_35_	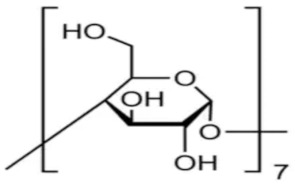	TCI *	7585-39-9	≥0.998
2-HP-β-cyclodextrin Sodium hydroxide micropills	C_66_H_112_O_42_ NaOH	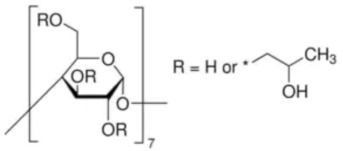	TCI * Avantor	128446-35-5 1310-73-2	≥0.998 ≥0.998

* TCI (Tokyo Chemical Industry, Tokyo, Japan).

## Data Availability

No applicable.

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
