# Peer review of "Conductometric Studies of Formation the Inclusion Complexes of Phenolic Acids with β-Cyclodextrin and 2-HP-β-Cyclodextrin in Aqueous Solutions"

_molecules, 2022, doi:10.3390/molecules28010292_

Round 1
Reviewer 1 Report
In this paper, the author studied the thermodynamics of host-guest interactions between β-cyclodextrin and several selected phenolic acid salts via conductometry. The paper is interesting and provides a number of useful parameters (K, ΔG, ΔS, ΔH). It is publishable subject to revision.
1.The first paragraph of the Introduction should be removed as it only repeats the information given in the abstract.
2.The author used a conductometry to determine the concentration of ions in the solution. Therefore, this method should be sufficiently introduced. Basic relations between the equilibrium constant and the molar conductivity should be provided.
3.Figures 1 and 3 provide identical data. Only Fig. 3 should be kept in the manuscript.
4.The thermodynamic functions ΔG, ΔS, ΔH are given with an extreme accuracy (8 digits, see Tables 6-9). The parameters are obtained from the temperature dependence of K. As such, they should be presented with the same precision. The parameters should be rounded to 4 digits.
5.The author only presents the data obtained and does not discuss them. It is necessary to compare the parameters obtained (K, ΔG, ΔS, ΔH) with other previously published studies if they are available.
6.A comparison is provided only with ref. [24] which is a self-citation. It is important to refer to the data obtained by other authors.
7.There are many analytical techniques to study the formation of the host-guest complexes. The author should discuss the advantages and limitations of conductometry.
8.The structure of the complexes should be suggested and discussed as well. Do you plan to conduct a spectral characterization of the complexes obtained?
Author Response
Answer for the Reviewer 1
I would like to thank you for the constructive comments and corrections of the manuscript to enhance its scientific values. In the revised version of my paper I included all comments and suggestions. I hope that the responses to the comments/suggestions presented below are sufficient and satisfactory. In the revised version of the work, all the changes are marked in yellow. Our responses to your comments are below.
- The first paragraph of the Introduction should be removed as it only repeats the information given in the abstract.
At the suggestion of the reviewer, the paragraph has been deleted.
- The author used a conductometry to determine the concentration of ions in the solution. Therefore, this method should be sufficiently introduced. Basic relations between the equilibrium constant and the molar conductivity should be provided.
Following the reviewer's suggestions, equations linking the basic relationships between the equilibrium constant and molar conductivity have been added.
- Figures 1 and 3 provide identical data. Only Fig. 3 should be kept in the manuscript.
At the suggestion of the Reviewer, Figure 3 has been removed from the manuscript.
- The thermodynamic functions ΔG, ΔS, ΔH are given with an extreme accuracy (8 digits, see Tables 6-9). The parameters are obtained from the temperature dependence of K. As such, they should be presented with the same precision. The parameters should be rounded to 4 digits.
As suggested by the Reviewer, the values in Tables 6-9 have been standardized from the point of view of the presented accuracy.
- The author only presents the data obtained and does not discuss them. It is necessary to compare the parameters obtained (K, ΔG, ΔS, ΔH) with other previously published studies if they are available.
and
- A comparison is provided only with ref. [24] which is a self-citation. It is important to refer to the data obtained by other authors.
In response to the comments in points 5-6 of the review about the lack of literature data and citing only my work [24], I must explain that:
I have not encountered in the literature when interpreting the results in the earlier work [24] and the currently presented values of conductivity and stability constants of the tested inclusion complexes obtained by conductometric methods. The experimental results and computational methods presented by me are a new research method for these complexes (which was not previously used by anyone using the presented equations). Let's hope that thanks to this work, a new research method will appear in the analysis of cyclodextrin inclusion complexes.
- There are many analytical techniques to study the formation of the host-guest complexes. The author should discuss the advantages and limitations of conductometry.
Thanks to the use of the conductometric method, we are able to very accurately determine the values of the inclusion complex formation constants as a function of temperature changes. Thanks to these measurements, we are able to determine the values of thermodynamic functions very quickly and with high accuracy without using other research methods, e.g. calorimetry.
- The structure of the complexes should be suggested and discussed as well. Do you plan to conduct a spectral characterization of the complexes obtained?
The results of conductometric measurements only allow to propose the stoichiometry of the forming inclusion complexes (which was done in the work). They do not allow structural proposals. In the near future, we are planning spectral studies and theoretical calculations, which will most likely allow us to formulate hypotheses regarding the structure of the inclusion complexes that are forming.
The revised version of the article includes corrections to the English language recommended by a native English speaker.
Reviewer 2 Report
Overall, the present manuscript is well designed, and the experiment data can explain the conclusion and the results are also interesting. However, there are several aspects of the manuscript that could be improved. This manuscript was recommended to be published after major revision.
1. In Line 15, what does the experimental data mean?
2. In “Material” section, Solvent information is missing.
3. In line 152, a brief description of the preparation process of sodium salts of the tested phenolic acids is required.
4. In “Characterization Methods” section, preparation of all samples should be placed prior to testing method.
5. The title of this manuscript is the study of conductometric studies of inclusion complex. However, in the abstract section, the results showed that the stability of inclusion complex increases with lowering of the molar mass of cyclodextrin and lowering the temperature. My question is “what is the relationship between the conductivity of inclusion complex and its stability ”.
Author Response
Answer for the Reviewer 2
I would like to thank you for the constructive comments and corrections of the manuscript to enhance its scientific values. In the revised version of my paper I included all comments and suggestions. I hope that the responses to the comments/suggestions presented below are sufficient and satisfactory. In the revised version of the work, all the changes are marked in yellow. Our responses to your comments are below.
- In Line 15, what does the experimental data mean?
In line 15, the notation representing limiting molar conductivity ( ) has been replaced by (LCDNaDod). Thank you for catching the editorial error.
- In “Material” section, Solvent information is missing.
As suggested by the Reviewer, a description of the solvent has been added to the article in the Materials chapter:
The water used for the measurements was distilled twice and then passed through an ion exchanger to obtain the best purity (Behr Laboor-Technik-Germany). Water was distilled twice (specific conductivity ~ 10-6 S‧cm-1).
- In line 152, a brief description of the preparation process of sodium salts of the tested phenolic acids is required.
As suggested by the Reviewer, a description of the prepared sodium salts of the tested phenolic acids was added to the manuscript.
„The studied salts were obtained by mixing the appropriate amounts of acid and aqueous sodium hydroxide solution in a stoichiometric ratio of 1:1. The mixture was then heated and stirred to dissolve the acid and evaporate the solvent. All analyzed sodium salts of acids were recrystallized twice from aqueous ethanol solutions. The salts obtained were washed with acetone and dried in a reduced pressure in a Büchi glass oven B-580 at T = 373.15 K until constant weight.”
- In “Characterization Methods” section, preparation of all samples should be placed prior to testing method.
As suggested by the Reviewer, the order in the text has been changed.
- The title of this manuscript is the study of conductometric studies of inclusion complex. However, in the abstract section, the results showed that the stability of inclusion complex increases with lowering of the molar mass of cyclodextrin and lowering the temperature. My question is “what is the relationship between the conductivity of inclusion complex and its stability ”.
The stability of the tested inclusion complexes affects the values of the measured conductances. The greater the stability of the complex, the smaller the amount of charge carriers in the tested solution, and thus the lower the conductivity. Similarly, lowering the temperature reduces the amount of free ions in the solution (increases the durability of the tested complexes) and reduces their mobility, and then we observe lower conductivity.
The revised version of the article includes corrections to the English language recommended by a native English speaker.
Round 2
Reviewer 1 Report
The author answered my previous comments only partially. The discussion of the results must be improved. The results must be compared with previously published studies of beta cyclodextrin-phenolic acid complexes by other authors.
1.The author only presents the data obtained and does not discuss them. A comparison is provided only with ref. [24] which is a self-citation. It is important to refer to the data obtained by other authors. The parameters obtained (K, ΔG, ΔS, ΔH) must be compared with previously published studies: https://dx.doi.org/10.1016/j.foodchem.2008.09.048, https://dx.doi.org/10.1002/elps.202200075, etc.
2.The conductometry has been used to study inclusion complexes of beta-cyclodextrin several times before. See http://dx.doi.org/10.1016/j.jpba.2014.02.022 for a review. One technique is rarely used. Since the author used only one technique, the advantages and limitations of this technique must be clearly mentioned. There are other analytical techniques to study the formation of the host-guest complexes that provide more information, e.g., spectral methods. The structure of the beta cyclodextrin-phenolic acid complexes should be discussed as well. If it has a 1:1 stoichiometry, as suggested in the manuscript, it should be supported by other methods and/or specific references. Do you plan to conduct a spectral characterization of the complexes obtained? If so, mention it clearly at the end of the discussion.
3.The discussion of the paper must be expanded. It is not enough to address the reviewer’s comments in the cover letter alone. The changes must be reflected in the manuscript itself. If the discussion is improved and references to previous studies are included, the paper can be accepted for publication.
Author Response
Reviewer 1
Thank you very much for taking the time to review my work and very valuable comments and suggestions regarding its content. I would also like to thank you very much for pointing out three publications related to the subject of my work.
Taking into account the comments and suggestions contained in the review, the following text was added to the final part of the work (before Conclusion):
“A review of literature data shows that there are many research techniques (spectroscopic techniques, electroanalytical techniques and separation techniques) allowing to assess the stability of cyclodextrin inclusion complexes with various types of compounds that have potential use as drugs [32]. As the authors of the works presented in the literature emphasize, none of the research techniques used separately allows for a full structural assessment of the forming complexes [32-34]. They only allow to determine the stoichiometry of the forming complexes. In the presented work, we proposed stoichiometry 1 : 1 for the studied complexes. A similar stoichiometry for chlorogenic and cinnamic acids with beta-cyclodextrin was proposed by Polewski et al. [33] based on a spectroscopic, thermodynamic and molecular modeling study. Similar conclusions can be drawn from the work of Plasson et al. [34]. In their work affinity capillary electrophoresis was used for the simultaneous measurement of the p?a values and of the binding constants relative to the encapsulation of naturally occurring phenolic acids (rosmarinic and caffeic acids) with cyclodextrin. It should be emphasized that in the cited papers [33 and 34], all thermodynamic functions describing the complexation processes (regardless of the research technique used) are negative, as in the case of the results presented by us. This indicates the same mechanism for the formation of inclusion complexes. The presented works (especially the work [32]) show how difficult it is to determine the exact stoichiometry and structure of the forming complexes.
The results of conductometric measurements presented in this paper only allow to propose the stoichiometry of the forming inclusion complexes. Conductometry is one of the most accurate research techniques that allows for conducting this type of research. The innovative computational methods introduced by us in the previous work [24] and used in this work allow for an accurate assessment of the complexation process taking place in the tested systems. Unfortunately, similarly to other research methods used in the literature [32], it does not allow for a full structural assessment of the forming inclusion complexes based on the results. Therefore, in order to supplement this information, we plan in the future spectroscopic and molecular modeling studies in the mixtures containing both a- and b-cyclodextrins.”
In the revised version of the work, all the changes are marked in yellow.
Reviewer 2 Report
None
Author Response
Thank you very much for taking the time to review my work and very valuable comments and suggestions regarding its content.